# Review of Crop Wild Relative Conservation and Use in West Asia and North Africa

**DOI:** 10.3390/plants13101343

**Published:** 2024-05-13

**Authors:** Nigel Maxted, Joana Magos Brehm, Khaled Abulaila, Mohammad Souheil Al-Zein, Zakaria Kehel, Mariana Yazbek

**Affiliations:** 1School of Biosciences, University of Birmingham, Edgbaston, Birmingham B15 2TT, UK; j.m.brehm@bham.ac.uk; 2National Agricultural Research Center, P.O. Box 639, Baq’a 19381, Jordan; kabulaila@gmail.com; 3Department of Biology and Natural History Museum, American University of Beirut, P.O. Box 11-0236, Beirut 1107-2020, Lebanon; ma73@aub.edu.lb; 4Genetic Resources Section, International Center for Agricultural Research in the Dry Areas, Beirut 1108-2010, Lebanon; z.kehel@cgiar.org (Z.K.); m.yazbek@cgiar.org (M.Y.)

**Keywords:** conservation, crop wild relative, CWR, food security, genetic erosion, plant genetic resources, prebreeding, threat assessment

## Abstract

Ensuring global food security in the face of climate change is critical to human survival. With a predicted human population of 9.6 billion in 2050 and the demand for food supplies expected to increase by 60% globally, but with a parallel potential reduction in crop production for wheat by 6.0%, rice by 3.2%, maize by 7.4%, and soybean by 3.1% by the end of the century, maintaining future food security will be a challenge. One potential solution is new climate-smart varieties created using the breadth of diversity inherent in crop wild relatives (CWRs). Yet CWRs are threatened, with 16–35% regarded as threatened and a significantly higher percentage suffering genetic erosion. Additionally, they are under-conserved, 95% requiring additional ex situ collections and less than 1% being actively conserved in situ; they also often grow naturally in disturbed habitats limiting standard conservation measures. The urgent requirement for active CWR conservation is widely recognized in the global policy context (Convention on Biological Diversity post-2020 Global Biodiversity Framework, UN Sustainable Development Goals, the FAO Second Global Plan of Action for PGRFA, and the FAO Framework for Action on Biodiversity for Food and Agriculture) and breeders highlight that the lack of CWR diversity is unnecessarily limiting crop improvement. CWRs are not spread evenly across the globe; they are focused in hotspots and the hottest region for CWR diversity is in West Asia and North Africa (WANA). The region has about 40% of global priority taxa and the top 17 countries with maximum numbers of CWR taxa per unit area are all in WANA. Therefore, improved CWR active conservation in WANA is not only a regional but a critical global priority. To assist in the achievement of this goal, we will review the following topics for CWRs in the WANA region: (1) conservation status, (2) community-based conservation, (3) threat status, (4) diversity use, (5) CURE—CWR hub: (ICARDA Centre of Excellence), and (6) recommendations for research priorities. The implementation of the recommendations is likely to significantly improve CWRs in situ and ex situ conservation and will potentially at least double the availability of the full breadth of CWR diversity found in WANA to breeders, and so enhance regional and global food and nutritional security.

## 1. Introduction

Crop breeders are currently faced with two over-riding, existential challenges: how to continue increasing crop production to feed the growing human population, while mitigating the expanding negative impact of climate change on agriculture. The human population is today 8.09 billion (30 January 2024) and is predicted to rise to 9.7 billion by 2050 [1]. To feed the global human population in 2050 we will require nutritious food supplies to increase by 60% globally [2], while climate change is predicted to reduce agricultural production by 2% each decade [3]. Global efforts to attain food and nutrition security against a backdrop of significant climatic, demographic, and socio-economic constraints depend on resilient agricultural production systems, underpinned by a genetically diverse portfolio of crops, crop varieties, and intra-varietal diversity, and greater varietal biotic resistances will reduce pesticide usage [4].

Tanksley and McCouch (1997) [5] reported that 95% of tomato genetic diversity is found in wild *Lycopersicon*/*Solanum* spp., not the crop itself. Harnessing the largely untapped potential of crop wild relatives (CWRs)—wild plant species that are the progenitors of crops and have increasing indirect use value as gene donors for current and future crop improvement [6,7]—can contribute to this diversity, helping to support the transition to fully sustainable agriculture, supporting a food and nutritionally secure future [8]. This is because in their natural habitats, CWR populations retain a wide range of adaptive abiotic, biotic, and agricultural traits such as drought tolerance, resistance to diseases, and increased yield.

The West Asia and North Africa (WANA) region comprises 20 countries (Algeria, Bahrain, Cyprus, Egypt, Iran, Iraq, Jordan, Palestine, Kuwait, Lebanon, Libya, Morocco, Oman, Qatar, Saudi Arabia, Syria, Tunisia, Turkey, the United Arab Emirates, and Yemen). West Asia, in particular, contains the fertile crescent which is the global hotspot of CWR diversity containing more than 40% of global CWR diversity [9]. WANA encompasses 2 of the 36 biodiversity hotspots (Mediterranean Basin and Caucasus) [10]; 2 of the 10 Vavilov centers of crop diversity (agrobiodiversity) (Asia Minor and Mediterranean [11,12]; and is the richest global center for CWR diversity (Figure 1a [9,13]). WANA is also the home to 45 of the proposed top 150 global sites for in situ CWR conservation (Figure 1b [13]). Turkey is the country with the highest number of native CWR taxa at the global level, whereas Lebanon is the country with the highest number of native CWR taxa for the country size, and the Hawash basin in Northwest Syria is the region with the highest number of CWRs per unit area [9]. Three hundred kilometers East of Hawash at Abu Hureyra in Syria, archaeologists found early proof for the development of the first agricultural crops by pre-Neolithic groups with rye cultivation dated to the Epi-Paleolithic era (c. 11,050 BCE) [14].

The ‘fertile crescent’ is the crescent shaped region to the North and West of the Arabian desert, in current-day Turkey, Syria, Lebanon, Iraq, Israel, Palestine, and Jordan. The CWR taxa occurring there provided the founding resources and impetus for the invention of agriculture by early farmers. Populations of these species were the progenitors of many of the earliest domesticated crops [15] and still grow wild in nature, often in close proximity to farmers’ fields in WANA. The region retains agricultural significance, not just because of its place in history, but also because it still contains wild populations of progenitors and genetically diverse landraces, although both local CWRs and landraces today are highly threatened with extinction or genetic erosion in their natural or anthropogenic habitats.

Specifically, WANA is home to vitally important CWR diversity [9] related to cereals (*Triticum monococcum* L., *T. durum* Desf., *T. turgidum* L., and *T. aestivum* L.—wheats; *Hordeum vulgare* L.—barley; *Secale cereale* L.—rye; *Avena byzantina* Koch.—red oat; *A. sativa* L.—oat); legumes (*Cicer arietinum* L.—chickpea; *Lathyrus ochrus* (L.) DC.—Cyprus vetch; *Lens culinaris* Medik—lentil; *Pisum sativum* L.—garden pea; *Vicia ervilia* (L.) Willd.—bitter vetch; *Vicia faba* L.—faba bean); oil-producing plants (*Olea europaea* L.—olive; *Linum* L., *Brassica* L., *Camelina* Crantz., and *Eruca* Mill.); melons (*Cucurbita* L. and *Cucumis* L.); vegetables (*Allium* L. spp.—onions; *Asparagus officinalis* L.—asparagus; *Beta vulgaris* L.—beets; *Brassica* spp.—cabbages; *Daucus carota* L.—carrot; *Eruca* spp.—rocket; *Lactuca sativa* L.—lettuce; *Valerianella locusta* (L.) Laterr.—cornsalad; *Lepidium sativum* L.—garden cress; *Pastinaca sativa* L.—parsnip; *Petroselinum crispum* (Mill.) Fuss—parsley; *Portulaca oleracea* L.—purslane; *Tragopogon porrifolius* L.—salsify); fruit crops (*Malus* Mill., *Pyrus* L., *Punica* L. *Ficus* L., *Cydonia* Mill., *Cerasus* Mill., *Amygdalus* L., *Vitis* L., and *Pistacia* L.); fodder and forage crops (*Hedysarum coronarium* L.—Italian sainfoin; *Lathyrus cicera* L.—chickling vetch; *Medicago sativa* L.—lucerne; *Trifolium resupinatum* L.—strawberry clover; *Trigonella foenum-graecum* L.—fenugreek; *Onobrychis viciifolia* Scop.—sainfoin; and *Vicia sativa* L.—common vetch); and dye plants (*Crocus sativus* L. and *Rubia tinctorum* L.) [11]; see some examples of CWRs in the region in Figure 2.

Despite the 7.4 million ex situ accessions held in ≈1750 genebanks globally (306,945 from WANA and 132,793 in ICARDA, 2008), several authors draw attention to the lack of sufficient access to diversity and note it is restricting plant breeding outcomes [16]. At the same time, CWRs are suffering erosion and extinction: 16 to 35% of CWRs are assessed as threatened in different regions ([17] for Europe; [18] for Meso-America). Global analysis of priority CWR conservation status found that about a third were unconserved, a third were poorly conserved (<10 accessions per CWR taxon), and 95% required additional collections [19]. It is well established that the conservation ideal is to apply in situ and ex situ techniques in a complementary manner [20]. In practice, however, the application of in situ conservation has been ignored with only a handful of active genetic reserves, either in protected areas or other effective area-based conservation measures (IUCN-WCPA Task Force on OECMs, 2019 [21]) [22], and even these do not meet the applicable conservation standard [23]. Despite the lack of CWR conservation efforts and the difficulty for breeders in obtaining CWR diversity, breeding using CWRs is increasing annually in terms of (a) crops where CWRs are used in breeding, (b) traits obtained from CWRs, and (c) the number of CWR taxa utilized [24].

The policy context is focused and explicit. The CBD Kunming-Montreal Global Biodiversity Framework [25] recognizes that “*genetic diversity helps species adapt to changing conditions, contributes to ecosystem resilience, and improves restoration success*”, yet “*genetic diversity is declining due to habitat loss, fragmentation, overharvesting, and other human activities*”. Therefore, its primary target is Goal A, “*The genetic diversity within populations of wild and domesticated species, is maintained, safeguarding their adaptive potential*”. While the UN sustainable development goals [1] highlighted the need to eradicate extreme poverty and hunger; Goals 1, 2, and 3, but particularly 2.5, aim to achieve the following: “*By 2020, maintain the genetic diversity of seeds, cultivated plants, and farmed and domesticated animals and their related wild species, including through soundly managed and diversified seed and plant banks at the national, regional, and international levels …*”. These goals, although focused and explicit, have not been met, suggesting the need for radical action beyond the status quo. Another indicator is the State of Food Security and Nutrition in the World report [26] which states that in 2022, between 691 and 783 million people faced hunger (281.6 million in Africa and 401.6 million in Asia). Globally, the figures for moderate to severe food insecurity have been rising since 2014 and nearly one in three people in the world (2.37 billion) did not have access to adequate food in 2022 and suffered moderate to severe malnourishment.

While there are no detailed estimates of the costs and benefits of traditional crop variety and CWR conservation, it is estimated that approximately 30% of the modern crop production increase is due to the use of CWR genetic diversity and that this has an annual value of approximately USD 115 billion worldwide [27], while more recently PWC (2013) [28] estimated that CWRs contribute USD 120 billion to increased crop productivity per annum for just 29 of the world’s priority crops. The increasing use of CWR diversity in crop improvement will help to mitigate the threat of genetic erosion and extinction facing this resource and sustain diversity retention. The globally important concentration of CWR diversity in the WANA region offers the foundation for an exceptional, paradigm changing opportunity to humankind to systematically conserve the breadth of crop-related diversity and ensure the maximum potential diversity is available for breeder exploitation. Taking this opportunity could at least double the diversity available to users in the WANA region, which, in turn, would generate substantial economic gain and further underpin the utilization of genetic resources in contributing to long-term food security in the region.

The International Center for Agricultural Research in the Dry Areas (ICARDA) is a CGIAR center and an international organization undertaking agricultural research to enhance livelihoods in temperate drylands in the face of global challenges such as climate change, food and nutritional insecurity and the agrobiodiversity crisis of reduced diversity. Conserving and using plant diversity to meet future climate- and market-related challenges is one of the five strategic research priorities for ICARDA. It thus plays a crucial global role in promoting the conservation and sustainable use of agrobiodiversity in the dry areas. Its genebank holds vital resources of unique collections of global significance. The collections total more than 155,000 plant accessions and 1380 strains of associated rhizobia of major winter cereals, food legumes, forage, and rangeland species drawn from the Vavilov centers of plant diversity (including the ‘fertile crescent’ in Western Asia, the Abyssinian highlands in Ethiopia, and the Nile Valley).

The aim of this paper is to signal a paradigm shift in WANA CWR conservation and use, this will include a review of (i) the current conservation of CWRs, (ii) the community-based CWR conservation, (iii) the threat status of CWRs, (iv) the usage of CWR diversity, (v) the ICARDA Centre of CWR Excellence, and (vi) recommendations for CWR research priorities.

## 2. Conservation Status

Successful CWR use in varietal improvement depends on whether the user community is able to effectively access the conserved CWRs, so that adaptive trait diversity can then be exploited. Increasingly, it is recognized that effective conservation is tied to the formulation of regional or national CWR conservation strategies and action plans [29]. This involves a series of key steps in conservation planning (see Figure 3). The essential first step in effective conservation is the creation of a CWR checklist, which is a comprehensive list of all CWR taxa, either found within a defined geographic area (floristic checklist) or taxonomic/gene pool grouping (taxonomic checklist) [30]. The checklist is commonly too large to be the basis of active conservation implementation, so the list is prioritized, possibly on the basis of closeness of relationship to the crop (ease of introgression), threat of genetic erosion/extinction, and economic value of related crop [31]. The priority taxa from the CWR inventory then may be elaborated with additional data to aid conservation implementation [30]. There has been no comprehensive CWR inventory produced for WANA, but individual inventories for West Asia (fertile crescent) of 220 CWR taxa and North Africa of 112 CWR taxa have been published ([32,33], respectively); there are also national CWR inventories for Cyprus [34], Iran [35], Israel [36], Jordan [37], Lebanon [38], Turkey [39], Tunisia [40], and for the Arabian Peninsula [41], as well as for WANA priority cereals [42] and legumes [43]. Combining the two regional lists for West Asia [32] and North Africa [33], together with those global priority CWRs found in WANA [9], we produced a WANA priority list of 488 CWR taxa (Appendix A [44]).

The taxon richness maps for CWRs in North Africa [33] and West Asia [32] are shown in Figure 4. These figures demonstrate that CWR diversity is not split evenly through WANA: in North Africa, it is focused on coastal Morocco, Algeria, and Tunisia; and in the fertile crescent, it is focused in West Asia. In terms of genetic diversity studies and conservation gap analysis, there are many studies that investigate the patterns of both within- and between-taxon CWR diversity, but there have been no broad WANA-wide studies of the breadth of CWR diversity with clear conservation goals and recommendations for in situ and ex situ actions.

### 2.1. In Situ Conservation

Having recognized the rudimentary nature of in situ CWR conservation in WANA, conservation planning studies on priority North African CWRs identified the top 10 potential sites for CWR in situ conservation (Figure 5) and locations are focused on coastal Morocco, Algeria, and Tunisia [45], while the top 10 proposed sites for CWR conservation in West Asia are first in the western fertile crescent and then across Turkey (Figure 5) [32]. These results and recommendations echo those established by the Global Environment Facility in situ conservation of CWR project on “*Conservation and Sustainable Use of Dryland Agrobiodiversity in the Fertile Crescent*” [46], and the Global Crop Diversity Trust’s “Towards a regional conservation strategy for PGR in WANA” analysis [47]. The former recommended both in situ and ex situ actions involving the monitoring of CWR richness, abundance, and frequency, as well as reduction in the factors associated with the degradation of their natural habitats. The project undertook ecogeographic and botanic surveys in 55 monitoring areas in the drylands and mountainous areas of Jordan, Lebanon, Palestine, and Syria between 2000 and 2010 [46]. The results indicated a continued CWR genetic erosion, mainly due to overgrazing and the destruction of natural habitats following land reclamation for urbanization and agricultural encroachment. Potential sites were identified and genetic reserves implemented that were hotspots with a large number of *Aegilops* and wild *Triticum* species [48,49]. Some of these sites are still monitored but the specific indicators that monitor the population levels and trends over time have not been rigorously analyzed nor individual sites’ management regimes revised accordingly.

There are a large number of priority CWR species in WANA that occur almost entirely in the disturbed habitats, like roadsides, formerly cultivated terrace and ruins of historical areas, as well as in agro-ecosystems, such as in cultivated fields, field margins, grassland, and orchards or on their boarders. Often CWRs are missing any kind of formal protection now and probably in the foreseeable future, as they occur in areas that are subject to direct human interventions and considered a target to be removed when local authorities upgrade or clean the roadsides from weedy plants that are assumed to spread fire when dry. For example, the Oneiza Triangle part of Shoback in Southern Jordan is a known CWR hotspot, but municipality workers were witnessed digging out bushes of *Noaea mucronata* (Forssk.) Asch. & Schweinf. (a common spiny species) that grow side by side with supposedly nationally protected *Lactuca aculeata* Boiss. and *L. orientalis* (Boiss.) Boiss. (a cultivated Roman lettuce CWR), eradicating the whole ecosystem and leaving no room for natural recovery. In another Jordanian example, on the campus of the University of Jordan, seeds were collected from a very large population of *L. aculeata* just before converting the whole area into a paved car parking. However, it should be noted that overall CWRs appear well-adapted to disturbed habitats and they grow exactly on the edges of the streets or cracks of the asphalt, making use of naturally evolved water-harvesting techniques.

Faced with cases of careless ‘weed clearance’, it is important to raise awareness and set legal frameworks, or maybe elaborate on existing ones, to rescue what remains of such a wealth of species. For example, in Jordan, Agriculture Law 13 of 2015 in Article 33 prohibits the cutting, burning, or trimming of wild plants that occur in the forests, and Article 34 applies the same prohibition on any wild rare plant in general. Article 34 also states that it is up to the Minister to establish a list of priority and high-value species for active conservation. However, there is no evidence of this law being enforced, even for species that are important for food security, such as CWRs! The fact that such laws exist means they could potentially be implemented to conserve CWR populations. This example is from Jordan, but other countries of the region (and further afield) have more or less similar situations that can be modified to help in implementing protection measures to conserve CWR diversity.

Although systematic CWR conservation should ideally apply both in situ and ex situ conservation measures [50], within WANA almost 100% of conservation activities are CWR population sampling of seeds for ex situ genebank storage. This is particularly concerning as in situ populations are under such extensive pressure from multiple threats, such as habitat destruction, urbanization, overgrazing, and climate change. More consistent in situ conservation is an urgent priority, especially as ex situ conservation cannot be used to secure the full breadth of CWR diversity.

### 2.2. Ex Situ Conservation

In terms of CWR ex situ conservation, population seed sampling and storage in a −18 °C genebank is currently the routine method of CWR conservation, and accessing seed samples from genebanks is the standard way users access germplasm [51]. There have been very few analyses for planning ex situ collection in WANA, but North African [45] and West Asia [32] grid cells where additional ex situ collection is required have been identified (Figure 6). In North Africa, coastal areas appear particularly under-collected, but it is clear there is a need for further collection and ex situ conservation throughout North Africa, while for West Asia, Western Turkey, southern coastal Turkey, Cyprus, and the fertile crescent are identified as regions for further collection and ex situ storage. It seems likely that all countries in the region would benefit from further collection, but Western Turkey and the northwestern element of the fertile crescent may prove particularly fruitful.

The global CWR project ‘Adapting Agriculture to Climate Change’ [52,53] analyzed the conservation status of global ex situ CWR collections for 1076 CWR taxa related to 81 crops, some genepools were adequately conserved (e.g., potatoes, eggplants, and rice), but overall globally 95% of CWR taxa are insufficiently represented, and a third of CWRs had no seed samples held ex situ [19] and further collection should be in the identified hotspots such as WANA. Novel collection should be based on distributional analysis and evidence of conservation gaps and re-collecting of known CWR locations should be avoided. In WANA, multiple global and national institutions actively contribute to the ex situ conservation of CWRs to ensure the continuing availability of crucial adaptive traits for crop improvement. The largest genebanks holding CWR diversity vary in their levels of capacities and include (a) the ICARDA Genebank (with the CWR collections maintained at Terbol, Lebanon) and with a main focus on wheat (*Triticum* and *Aegilops* spp.), barley (*Hordeum* spp.), lentils (*Lens* spp.), and forage legumes (*Lathyrus*, *Medicago*, *Trifolium,* and *Vicia* spp.); (b) the Leibniz Institute of Plant Genetics and Crop Plant Research (IPK) which, while located in Germany, maintains a substantial collection of CWRs from the WANA region, mainly focusing on cereals and legumes CWR; and (c) the Millennium Seed Bank based in the UK which focuses on the breadth of CWR diversity found in the region. Each of the countries in the region also has at least one national genebank for ex situ agrobiodiversity collections, but the technical expertise required to locate and identify CWR taxa means national genebanks have focused largely on collecting crop germplasm itself rather than native CWR diversity.

Accessing information about the holdings of international genebanks is relatively straightforward, especially when they publicly share their data through open-access platforms like Genesys (https://www.genesys-pgr.org). However, understanding the status of national genebanks presents more challenges for several reasons. Firstly, most of these national genebanks do not make their holdings data publicly available, either on their institutional portals or other public platforms. Secondly, many of these genebanks employ outdated data management systems, making it difficult to query holdings and identify any existing gaps. Moreover, the quality of the samples conserved, such as their viability, health, and maintenance of genetic integrity, remains uncertain. It is important to note that many early ICARDA collection missions targeted CWRs native to the WANA region, particularly of field crops and forage legumes. One single mission for field and forage legumes collected over 1800 seed and rhizobia samples and discovered several new CWR species (*Vicia kalakhensis* Khattab, Maxted & F.A.Bisby, *V. eristalioides* Maxted, and *Lathyrus belinensis* Maxted & Goyder) over the spring of 1986 in Syria and 1987 in Turkey [54]. Many of the collecting expeditions involving ICARDA staff have recently involved the repatriation of ICARDA-maintained collections to the country of origin.

Moreover, collaborative projects between national genebanks and international organizations that emphasize the utilization of the CWR germplasm in crop improvement result in germplasm being conserved nationally and with a safety duplicate deposited in one international organization. For example, in 2012, the National Agricultural Research Center of Jordan (NARC) and the Institute of Plant Genetics and Crop Plant Research (IPK) collaborated to collect barley seeds (and corresponding herbarium specimens) which were then shared between the two institutions. Additionally, NARC and the Center of Genetic Resources of the Netherlands (CGN) also collaborated in the collection of lettuce wild relatives. Most national collections in WANA are “safely duplicated” in other organizations and countries in the WANA region. Figure 7 indicates the number of CWR accessions conserved at the ICARDA genebank collected from the countries of WANA and Figure 8 globally. It should be emphasized that an understanding of the ex situ CWR collection and storage at national genebanks can only be achieved through capacity development, focusing on fundamental aspects of germplasm management, with a particular emphasis on documentation and data management to build the skill set required to sustain regional CWR diversity.

## 3. Community-Based Conservation

CWRs are a common component of the flora of the region and a known major source of stress tolerance genes transferable to key crop species responsible for global food security. However, the utilization of such precious resources in breeding programs is minor at the local level, which has been reflected in dealing with and prioritizing the protection of such species. On the contrary, in many cases they are dealt with as sprawling weeds growing here and there close to their cultivated relatives or on the field borders such as *Hordeum spontaneum* K. Koch, or in highly disturbed habitats, in the neglected areas and roadsides, such as relatives of lettuce, carrot, and some barley relatives, where no protection measure could be applied. Recently, there are many initiatives and projects that promote and support the implementation of the concept of biodiversity-sound agriculture systems such as permaculture. Such implementation is being gradually mainstreamed in the region as part of the package of approaches to face the effects of climate change; such systems would leave a space, sometimes known as an ecological corridor, for the associated biodiversity to take place in the vicinity of the cultivated land. In addition, there is considerable momentum lead by the FAO, mainly represented by its Commission for Genetic Resources for Food and Agriculture (CGRFA) in coordination with other related fora such as the CBD secretariat in monitoring and taking actions in the member countries.

Some CWRs are also wild edible foods such as *Asparagus* spp., *Lactuca* spp. and *Pistacia* spp., *Pyrus syriaca* Boiss., *Allium* spp., and others. They are harvested from the wild and utilized in different ways, but primarily as wild food plants. Anecdotal observation from the authors suggests that wild food plant harvesting is too intensive and threatens their populations. In special areas of conservation, where the local communities have a major role in conserving the whole ecosystem, CWRs and their habitats are conserved and sustainably utilized as part of the ecosystem conservation holistic approach. For example, in pastoral communities, there is a traditionally well-known approach of rangeland conservation called *Al Hima*, where grazing management by Bedouin tribes is rotated during the course of the season to allow the grazed area to recover in part [55]. However, in recent years herd sizes have expanded and the *Al Hima* system has not sustained production, in spite of all the efforts to revive it [56].

To protect this major pillar of plant genetic resources for PGRFA, there is a crucial need to raise the awareness of the local communities regarding such wealth surrounding them and the best way to sustainably utilize it for their direct use and grazing animals. There is also a vital need for a strong collaboration between genebanks and local communities in strengthening in situ conservation approaches and to broadly mainstream the utilization of biodiversity friendly approaches including revitalizing using local landraces that are well-adapted and need less harmful inputs that leave a greater chance for associated biodiversity, including CWRs, to occur in and surrounding the agroecosystem. On the other hand, the protected area authorities need to be educated and to have major roles in educating the indigenous communities in their protected areas. In addition, there is a need to recognize the benefit sharing of the local communities upon the collection and utilization of CWRs and their traditional knowledge and enable local communities to know their rights in the benefits created by their utilization.

## 4. Threat Status

The threat status of CWRs in the WANA region is largely unknown as there is no regional Red List for WANA. However, an analysis of the threat status of the plant diversity that occurs in any of the countries of the WANA region, as per the IUCN Red List of Threatened Species (https://www.iucnredlist.org/) [44], identified a total of 2351 plant taxa that have been assessed either at the global or at the regional level (Europe, Mediterranean, northern Africa, southern Africa or Pan-Africa), of which 29.0% (681 taxa) are CWRs of human or animal food crops (Appendix A [44]). Out of the 681 CWR taxa that have been assessed, 15.7% (107) are threatened [26 Critically Endangered, 47 Endangered, 34 Vulnerable] (Table 1), as well as 6.5% (44) which are Near Threatened (or Lower Risk), 68.0% (463) Least Concern, and 9.8% (67) Data Deficient. The genus *Lathyrus* includes the highest number of threatened wild relatives with 18 taxa, followed by *Allium* and *Astragalus* with 16 taxa each (Appendix A [44]). The genus *Lathyrus* also includes the highest number of Near Threatened taxa (9), followed by *Allium* (5), and *Carex*, *Pistacia*, and *Pyrus* with three each. Care needs to be taken when interpreting these results, i.e., just because *Lathyrus* is the genus with the highest number of threatened and Near Threatened taxa, it does not mean that they are the most threatened genera in the region. This may be because a high number of taxa were effectively assessed (66 for *Lathyrus*), followed by *Allium* (56) and *Cyperus* (47) but only with 16 and 1 threatened CWRs, respectively. Out of the 681 food-related CWR taxa assessed, 21.6% (147) priority CWRs in the WANA region (Appendix A [44]) have been assessed, of which 7.5% (11) are threatened (three Critically Endangered, five Endangered, three Vulnerable), 4.8% (7) are Near Threatened (or Lower Risk), 76.9% (113) Least Concern, and 10.9% (16) Data Deficient (Table 1). Further analysis identified at least 12.5% (85 species) of the assessed CWRs have their populations in decline, whereas 45.5% (310) are stable, 1.8% (12) are increasing, and the remaining 40.2% (274) are unknown. Similarly, 10.1% (15) of the priority CWRs have populations in decline, 56.1% (83) are stable, 1.4% (2) are increasing, and the remaining 31.8% (47) are unknown (Appendix A [44]).

Thirty-seven threats were identified for 42.7% (291 taxa) of the CWRs that occur in the WANA region (Figure 9). The top three main threats are (i) livestock farming and ranching, which affects 174 CWRs (of which 79 are threatened); (ii) housing and urban development, which affects 95 species (39 threatened); and (iii) agriculture practices, affecting 73 CWRs (24 threatened). Similarly, priority CWRs in the WANA region are mostly affected by (i) livestock farming and ranching, which affects 43 CWRs (11 threatened); (ii) housing and urban development which affects 17 species (3 threatened); and (iii) both tourism and recreation, and agriculture practices, each affecting 16 CWR species, of which no threatened species is affected by tourism and recreation and 6 are threatened by agricultural practices (Appendix A [44]).

## 5. Diversity Use

The justification for CWR conservation is that the germplasm has a high commercial and potential utilization value as a gene/trait donor to crops, along with its other functional and intrinsic ecosystem services value associated with any wild plant species [22]. Therefore, the predominant reason for active CWR conservation is not conservation itself but is associated with their use in breeding novel varieties. However, many breeders are reluctant to use CWR germplasm directly because of the linkage drag and the introduction of deleterious traits along with those desired [57]. Thus, prebreeding, involving the provision of pre-bred lines that already contain beneficial CWR traits for farmer and breeder usage [57,58], has increased the ease of access to CWR germplasm [52,59], and rapid advances in gene editing [60,61] are making linkage drag less of a limitation and reducing breeders’ reluctance to use CWR diversity to maintain food security which can be less readily justified. Therefore, the increasing use of CWRs can underpin food security under the changing climate and sustainable development. This is also relevant for forages which are consumed by livestock, as they are an important component of human food security.

Historically, CWR use in crop breeding goes back as far as modern crop breeding, i.e., back to the 18th century; that is of course not considering the use of the early agriculturist of wild progenitors and crops to domesticate the crops we know now in the first place. There are no known crops (yet) that do not have genes from CWRs, no matter how changed they are through domestication and more modern improvement tools. More recently, the use of CWRs in breeding efforts gained prominence in the 1970s and 1980s. Wild species have contributed traits like disease resistance, drought tolerance, and a wide array of other adaptive traits that improve crop varieties [8]. This has its own challenges though, first and foremost the identification of the genotype with the desired trait, the genes responsible for the desired trait and to be introgressed into breeding parental germplasm (gene discovery), and dealing with linkage drag and interspecific crossability, hindering the incorporation of valuable traits from wild species into cultivated crops. The first step in using CWRs by plant breeding is to access the information on these accessions (phenotypic and genomics). There is a little trait information about CWRs [62] and enriching genetic resources collections with trait information for breeding is resource consuming [63], since one needs to identify the needed trait to evaluate, especially looking at the size of the collection and the presence of genotype-by-environment interactions [64].

Various techniques are being developed to overcome challenges to CWR usage. For the challenging gene discovery in extensive collections, including CWRs, a combination of ecogeographic methods with machine learning can be used. It involves probing specific sets of germplasm, like core collections and composite collections. One such way of creating subsets is through the focused identification of germplasm strategy (FIGS) which uses environmental data from collection sites to model trait–environment correlations, resulting in a subset of germplasm that is more likely to carry the desired adaptive traits [65,66,67]. FIGS has proven its worth in identifying genetic resources with traits like disease resistance and stress adaptation in crops like wheat, barley, faba bean, and soybean [66,68,69,70]. In a study focusing on harnessing the potential of CWRs for developing future crops through breeding, Bohra et al. (2022) [71] suggested that new tools would play an important role in accelerating access to CWR adaptive trait diversity for breeding applications. These include emerging breeding tools like genomic selection and optimum contribution selection which are increasingly playing a role in identifying the most favorable combinations of desired alleles in crosses between exotic by elite varieties and precise gene-editing tools that hold significant promise for minimizing linkage drag [71]. An overview of challenges and bottlenecks of CWR use, and the advancements made to tackle these challenges, are reviewed in a study by Dempewolf et al. (2017) [57]. An extensive survey of the literature in the past decade, conducted by the authors, including 127 different crops and 970 CWR taxa recorded 4157 potential or confirmed “uses” of CWRs in crop improvement, which were grouped into seven “breeding use classes” as follows: agronomic trait (485), abiotic stress (700), biotic stress (2427), fertility trait (272), morphological trait (20), phenological trait (54), and quality trait (199).

After sunflower, the crops with the second most extensive breeding uses of CWRs documented was wheat (*Triticum aestivum* L.) with CWR material obtained from the WANA region. In barley, wild relatives have been extensively employed in barley breeding, particularly utilizing *H. spontaneum* and *H. bulbosum* L. The desired traits encompassed disease resistance, targeting traits such as net blotch, scald, leaf rust, and powdery mildew. Ongoing efforts of the authors include an extended exploration of diverse attributes like the photothermal response, protein content, high beta glucans, and cold tolerance [72]. The incorporation of CWRs in food legumes has also proven effective in enhancing adaptive traits and varietal improvement. Lentil (*Lens culinaris*) breeding has seen advancements in *Sitona* weevil and *Anthracnose* resistance, *Fusarium* wilt, salinity tolerance, earliness, enhancing iron and zinc contents, as well as *Orobanche* resistance [73]. Chickpea (*Cicer arietinum*) improvement strategies have focused on resisting *Botrytis* grey mold, root knot nematode, and addressing water-logging challenges, tolerance to cold, heat, and drought, along with improved micronutrient content (Iron, Zinc, Selenium, etc.), to name a few. Furthermore, in grass pea (*Lathyrus sativus* L.), a climate-smart crop with established drought resistance, the wild species *Lathyrus cicera* L., *L. blephicarpus* Boiss., *L. heirosolymitanus* L., *L. annuus* L., *L. cassius* Boiss., and *L. gorgoni* Parl. have contributed to traits such as tolerance to drought, heat, and water-logging, in addition to reducing ODAP, a neurotoxin that can lead to paralysis in the lower limbs. It is important to highlight that the use of CWRs originating from the WANA region is not restricted to users from the region, these resources are a truly global priority resource, as indicated in Figure 10 by the distribution of CWR accessions from the ICARDA collection.

Addressing challenges posed to the use of CWRs in prebreeding requires improved data management and enhanced collaboration between research communities, most importantly genebank managers and curators from one side and breeders on the other side, and the increased investment in prebreeding programs as a bridge along the breeding pipeline continuum. Beyond their significance in crop improvement, the conservation of CWRs is essential for conserving biodiversity at large, promoting ecosystem health, and ensuring the long-term sustainability of plant communities in diverse landscapes. CWRs are also just wild plant species and integral components of plant communities, playing a crucial role in fostering the health and balance of rangelands and ecosystems, contributing to the overall biodiversity and ecological resilience of these environments. This is an area that is largely overlooked both in WANA and globally. It is also interesting to note the variety of purposes and research associated with the use of CWRs from the ICARDA genebank. Figure 11 indicates the diversity of uses (a) and user types (b) for which the CWR materials originating in the WANA region are being requested and distributed.

## 6. CURE—CWR Hub (ICARDA Centre of Excellence)

Despite the establishment of the dual ICARDA genebanks, which were split between two locations at Rabat, Morocco, and Terbol, Lebanon, 10 years ago, work on PGRFA conservation and use picked up very fast and the Terbol, Lebanon, center is being positioning as ICARDA’s hub for conservation, use, research, and education for crop wild relatives (CURE—CWR hub). Suitably located at the heart of the global center of agrobiodiversity, the research station at Terbol, Lebanon, is dedicated and focused on conserving and using the genetic diversity found in wild plant species related to major world crops (wheat, barley, faba bean, lentils, and chickpeas). It is envisaged that it will make a very significant contribution to ensuring global food and nutritional security, sustainable international agricultural systems, and achieving global policy requirements. It is also serving as a platform to strengthen research, education, and outreach, bringing together scientists, students, farmers, and other stakeholders to advance our understanding of CWR diversity, actively conserve its breadth, and maximize its sustainable uses. ICARDA’s unique collection is one of the most important assets in the CURE—CWR hub. This is a collection that was compiled over 40 years of collection missions in the dry areas, mostly focusing on the WANA region, guided by the natural distribution of CWR diversity and native species of ICARDA’s mandate crop (Figure 12).

In addition to the unique collection, CURE—CWR hub manages various facilities for the regeneration and conservation of the diverse annual and perennial CWR species, maintained in the ICARDA collection, each with its diverse requirements (i.e., self-pollinated, self-compatible cross-pollinated, self-incompatible cross-pollinated, shattering, etc.). The existing facilities include an open field, plastic houses, insect and pollen proof isolation cages (mesh), a rich herbarium collection emphasizing CWR diversity from the Central West Asia and North Africa region, and laboratories (for monitoring viability, seed health, and nutritional quality). The hub’s infrastructure will enable researchers to utilize CWR mostly in, but not limited to, prebreeding. The large diversity conserved can and is being used in restoration efforts (reintroduction of native species to degraded ecosystems) as well as in more basic fields of research (physiology, taxonomy, phylogenetics, etc.).

Possibly the most important aspect of the CWR—CURE hub is its commitment to sharing knowledge, tools, and resources with communities and organizations working to conserve and utilize CWR in WANA nationally and in other global regions. The hub provides a platform for training the next generation of plant scientists and agriculturalists working on all aspects of CWR conservation and use, enabling students and early-career professionals to gain hands-on experience in conservation, taxonomy, prebreeding, and related fields. Through group workshops and trainings, and degree and non-degree long-term individual education programs, the CWR—CURE hub is playing a pivotal role in building a community of experts and advocates for CWR around the world, promoting harnessing the potential of wild plant species to create a more resilient and equitable food system for future generations.

## 7. Discussion

WANA is the global region with the highest number of CWR taxa and the highest number per unit area. Consequently, there is a pressing need for national, regional, and global level efforts for CWR diversity conservation and enhanced utilization. This wealth of CWR diversity has the potential to help in sustaining production levels in the face of diverse challenges, including natural shocks such as extreme climate events, socio-economic disruptions, and geopolitical tensions such as reduced agricultural workforce, civil unrest, market volatility, food price spikes, conflict, and trade wars. By improving agricultural systems’ resilience, promoting agricultural diversification, and mitigating harmful production practices, CWRs can play an enhanced role in ensuring food and nutritional security. Most obviously, CWRs have the capacity to enhance crop and varietal resilience to the ever-evolving biotic and abiotic threats including those exacerbated with climate change.

The review concludes that there still exists a significant taxonomic and genetic diversity of CWRs in the WANA region. However, conservation efforts have been widely ad hoc, lacking strategic or systematic planning—this necessitates an urgent move to a more focused and coordinated approach, especially considering that these invaluable resources are increasingly threatened in the wild. While regional and national genebanks hold substantial ex situ collections of CWR diversity, the status of germplasm health and their viability, genetic integrity, taxonomic and genetic diversity, as well as their effectiveness as a representation of the full diversity found in nature remains unassessed. It is also worth noting that there is an evident, inexplicable, and unjustifiable lack of active in situ conservation of CWRs in WANA. The Global Environment Facility CWR project, implemented two decades ago in Syria, Lebanon, Jordan, and Palestine, and which focused on “*Conservation and Sustainable Use of Dryland Agrobiodiversity in the Fertile Crescent*” identified and established genetic reserves, but subsequently these sites have lacked active management to maintain CWR populations, despite some efforts to continue ad hoc monitoring. Implementing active management plans and establishing links between the conserved resources and their utilization could significantly increase the diversity available to breeders, thus strengthening the foundation for food security in the WANA region. Given the availability of tools and guidelines for in situ CWR conservation, it would be remiss not to apply them in the center of CWR diversity. Equally critical is the extension of community-based conservation efforts, which could include the application of OECM methods with the involvement of communities or individual farmers or land owners [23], but encourage the application of exemplar models of the ideal approach to be taken in WANA which would need to be developed.

The threat status of CWR diversity in the WANA region is poorly understood. Although 681 CWR species occurring in the WANA region have been assessed at the global level, only 21.6% are priorities to the region, i.e., 30.1% of total number of priority CWR have been assessed. This figure highlights the need to assess the level of threat for the remaining 69.9% of CWR diversity identified as regional priorities, many of which are potentially extremely useful for crop improvement. If the observed global trends among the CWRs apply in WANA, then we could approximate that around 15% of these priority CWRs are threatened. Adding the Near Threatened species, which may possibly become threatened if threats are not reverted, could raise the percentage of threatened priority CWRs to around 20% that may soon become extinct. It would also be beneficial to assess the CWR threat at the regional level within WANA. Developing a regional Red List would serve as an important instrument for conservation planners, implementers, and policy makers. The analysis of the globally assessed CWR taxa has identified major threats affecting CWR diversity throughout their distribution such as livestock farming and ranching, urban development, agriculture practices, and tourism and recreation. It is likely that these threats are also important threats affecting CWR in the WANA region. Notably, the “*Conservation and Sustainable Use of Dryland Agrobiodiversity in the Fertile Crescent*” project highlighted overgrazing and the destruction of natural habitats following land reclamation for urbanization and intensive agricultural encroachment [46] as major concerns in WANA.

Our review of the use of CWRs originating in the WANA region highlights their significance and potential utility in varietal improvement by providing valuable genes and traits to crops. However, current use is largely restricted to certain high-value crops and, apart from the efforts by ICARDA and very few national partners, occurs predominantly outside of the WANA region itself. Although in a sense it is irrelevant whether the CWRs are utilized within the region or not, the improved crop varieties or pre-bred lines incorporating CWR traits will naturally flow in the region through standard germplasm channels. ICARDA breeders make their new bred or pre-bred lines available to a wider community of breeders worldwide and specifically in the WANA region, which is the heart of the dry lands. Nevertheless, increasing the routine use of CWR germplasm or pre-bred line by national breeders would undoubtedly enhance their use, promote an appreciation of CWR diversity, and strengthen conservation efforts.

The formal establishment of the CURE—CWR hub at the ICARDA field station in Lebanon offers significant potential to advance all aspects of CWR conservation and utilization. It particularly serves as a focal point for research, education, and outreach, bringing together scientists, students, farmers, and other stakeholders, to advance our understanding of CWR diversity, actively conserve its breadth, and enhance its sustainable utilization. The CURE—CWR hub is therefore uniquely positioned to develop and refine the essential tools, guidelines, and exemplary case studies for the region.


**Recommendations for research priorities:**


Based on the review, the following recommendations are proposed in the WANA region:To establish a WANA-CWR expert group to coordinate CWR research and capacity building efforts, strategically positioning CWR in both national and regional agendas.Developing a regional CWR Conservation and Use Strategy and Action Plan to systematically address in situ (genetic reserve and OECM) and ex situ (seed banking) conservation in a coordinated manner throughout the region, considering prior research such as the sub-regional analyses of Zair et al. (2017) [32] and Lala (2017) [45] and using the available conservation planning tools [74] and associated templates provided by Magos Brehm et al. (2017) [29] and Dulloo et al. (2017) [75].To formulate, for each WANA country, a national CWR Conservation and Use Strategy and Action Plan for systematic in situ (genetic reserve and OECM) and ex situ (seed banking) conservation for their country using the *Interactive Toolkit for CWR Conservation Planning* [30], the available conservation planning tools [74], and the associated templates provided by Magos Brehm et al. (2017) [29] and Dulloo et al. (2017) [75], aligning with regional strategies, and possibly in collaboration with the CWR—CURE hub (ICARDA).To conduct a detailed assessment of current conservation action, at the taxonomic and genetic levels, is required to accurately develop conservation strategies and action plans. Although we have representative information on the CWR taxa held in regional and national genebanks, we have very little information on what percentage of CWR diversity found in nature is sampled and represented in genebanks; research should identify these gaps and attempt at filling them to ensure maximum coverage.To assess the threat status of CWRs at the global level for WANA regional endemics but also to assess the threat status of the remaining CWRs at the regional level, possibly first targeting the WANA priority CWRs identified in Appendix A [44].To identify and take mitigation actions against major threats impacting CWR diversity (livestock farming and ranching, housing and urban development, agriculture practices, and tourism and recreation) through input to inform the regional and national CWR Conservation and Use Strategy and Action Plans.To update regional CWR gap analysis using data collected for national and regional threat assessment and use this analysis to inform the regional and national CWR Conservation and Use Strategy and Action Plans to aid conservation planners, implementers, and policy makers.To identify key CWR populations of regional and national importance in diverse habitats to act as exemplary long-term monitoring sites/areas for the assessment of regional and national CWR threats and trends in CWR diversity and health status.To identify key CWR diversity that has global, WANA, and national utilization value in crop improvement and ensure the fact that such germplasm is conserved and made available to the three geographically distinct sets of users. Ideally, the germplasm would be characterized and evaluated so breeders are better able to select the most desirable samples for crop improvement.To develop or adopt an online information system where conserved CWR data and knowledge can be shared and made publicly available would undoubtedly enhance utilization. An example of such an approach is EURISCO{ XE “EURISCO” } (www.ecpgr.cgiar.org/resources/germplasm-databases/eurisco-catalogue/, accessed on 24 January 2024), Europe’s unique online database of actively conserved agrobiodiversity, including both ex situ and in situ holdings of CWR diversity.To formulate legal regional frameworks for the protection, monitoring, or characterization of CWRs found in the urban, protected areas or on-farm, aiming to provide more formal conservation of CWR diversity and raise biodiversity community awareness of CWR value. This should be performed by incorporating existing mechanisms for biodiversity maintenance in each country in WANA.To strengthen the collaboration and coordination between authorities that deal with the conservation of PGR and those of protected areas. This includes implementing measures of active conservation of CWR populations and the utilization of technical tools to describe and carry out optimal conservation of CWRs such as the *Descriptors for Crop Wild Relatives Conserved* In Situ [76] issued by the International Treaty of the FAO.To increase awareness about the concept of CWRs and their importance for local, regional, and global food security amongst academic and research institutes, decision makers, local communities, and the target users, the crop breeders. Ahead of such awareness, there should be an effort to identify which priority CWR species are found in each country.To develop CWR distribution maps, using data of national or overseas herbaria and seed collections, and adding their distribution’s future projection under different scenarios of climate change; such maps need to be kept updated as much as possible using field data.To invest in capacity building of national conservation staff to ensure they have the necessary taxonomist skill set to help CWRs in the wild and so be able to collect and conserve them ex situ and monitor their in situ populations.

## Figures and Tables

**Figure 1 plants-13-01343-f001:**
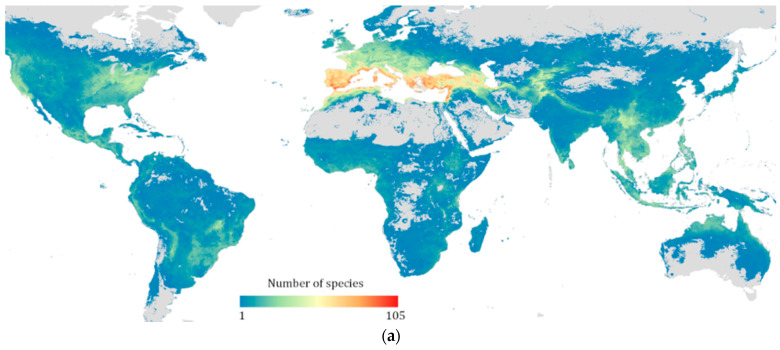
(**a**). Taxon richness for 1261 priority CWRs related to 167 globally important crops [13]; (**b**). Top 45 sites for global in situ CWR conservation located in the WANA region [13].

**Figure 2 plants-13-01343-f002:**
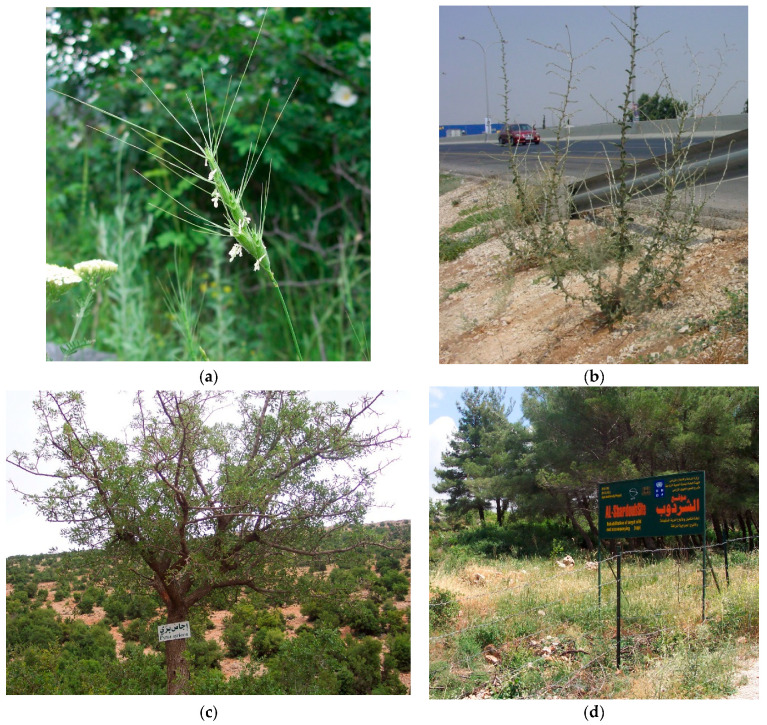
(**a**) *Aegilops triuncialis* L. growing on a field margin in Meghri, Armenia (Photo: N. Maxted), (**b**) *Lactuca aculeata* Boiss. & Kotschy growing on the roadside of the airport highway in Amman, Jordan (Photo: K. Abulaila), (**c**) *Pyrus syriaca* Boiss. growing in rough pasture in Suweida, Syria (Photo: N. Maxted), and (**d**) a genetic reserve established in Latakia province, Syria, in 2016 (Photo: N. Maxted).

**Figure 3 plants-13-01343-f003:**
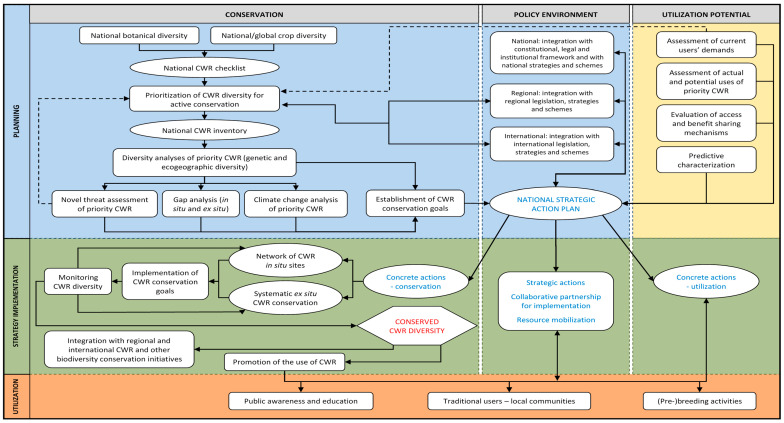
Development of strategic action plans for sustainable conservation and use of CWR diversity (adapted from Magos Brehm et al., 2017a [30]).

**Figure 4 plants-13-01343-f004:**
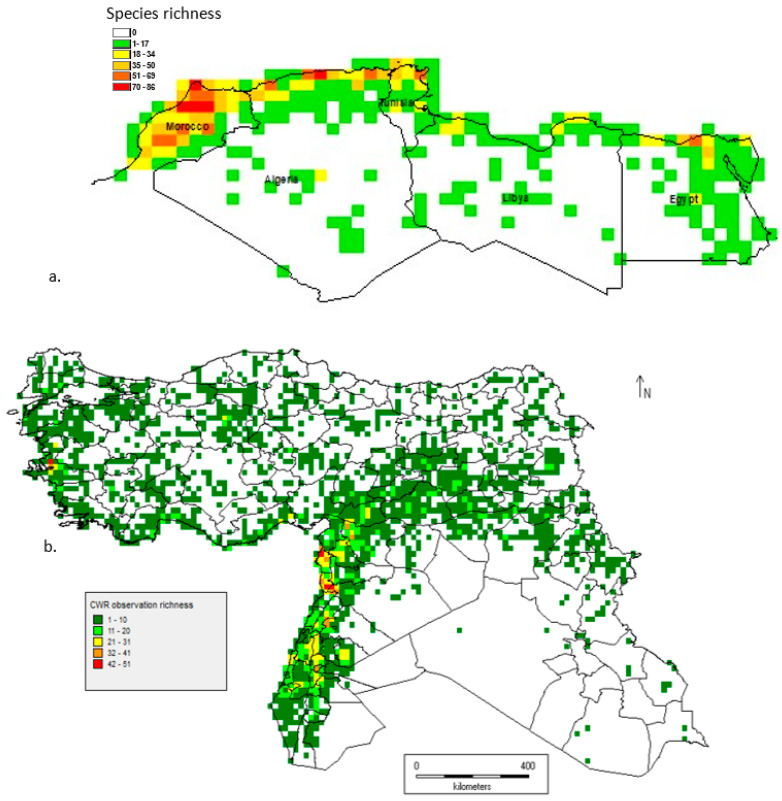
(**a**). CWR taxon richness in the North Africa region [33]; (**b**). CWR taxon richness in the West Asia region [32].

**Figure 5 plants-13-01343-f005:**
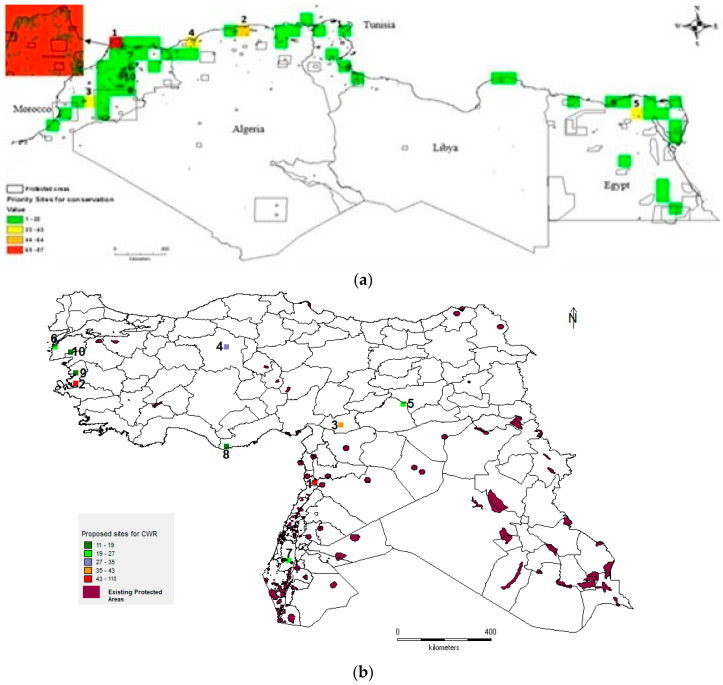
(**a**). Top 10 complimentary areas (1002 km cell size) for CWR in situ conservation in North Africa [33]; (**b**). Top 10 complimentary areas (1002 km cell size) for CWR in situ conservation in West Asia [32].

**Figure 6 plants-13-01343-f006:**
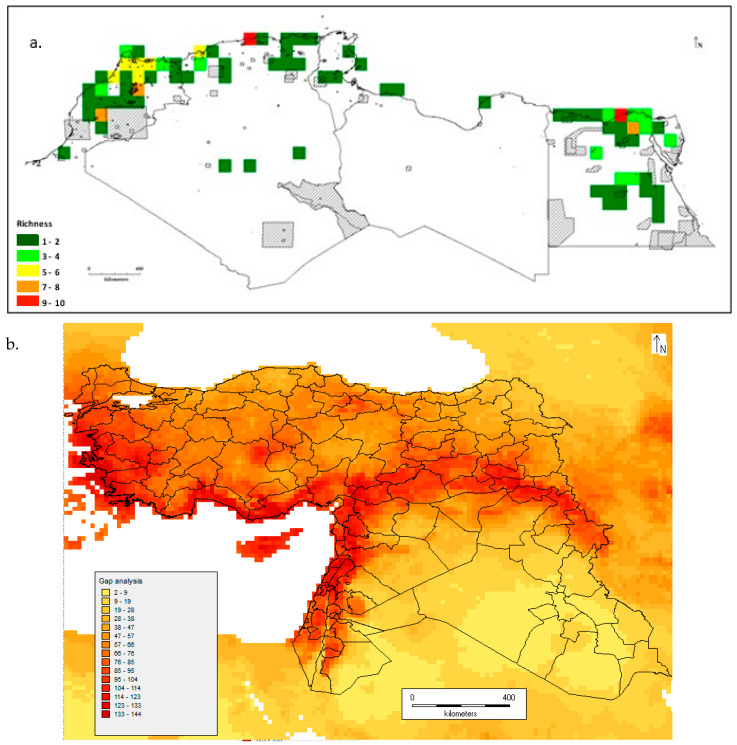
(**a**) Areas of CWR diversity in North Africa not conserved in regional and national genebanks [45]; (**b**) Areas of CWR diversity in West Asia not conserved in regional and national genebanks (grid cell of 5 min (~9 km^2^)) [32]; the legend displays the number of CWR taxa expected to exist in a single grid square.

**Figure 7 plants-13-01343-f007:**
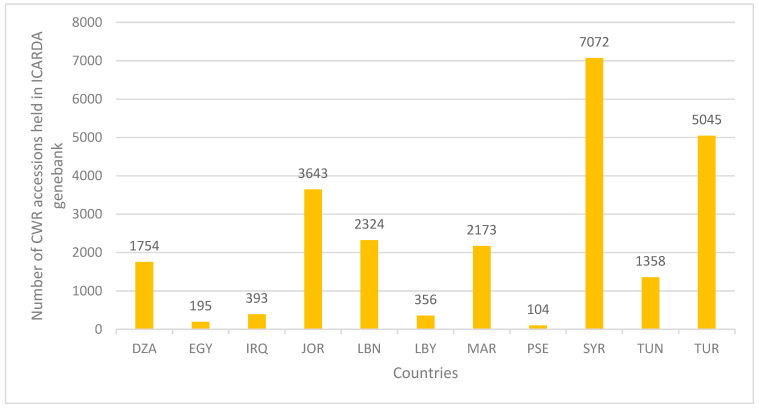
Number of CWR accessions conserved in the ICARDA genebank collected from WANA (DZA: Algeria; EGY: Egypt; IRQ: Iraq; JOR: Jordan; LBN: Lebanon; LBY: Libya; MAR: Morocco; PSE: Palestine; SYR: Syria; TUN: Tunisia; TUR: Turkey).

**Figure 8 plants-13-01343-f008:**
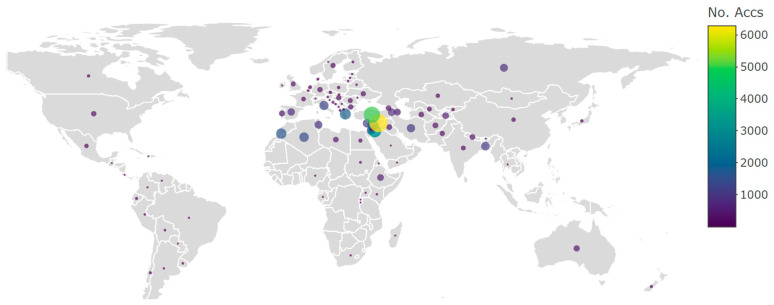
Number of CWR accessions conserved in the ICARDA genebank collected from countries globally.

**Figure 9 plants-13-01343-f009:**
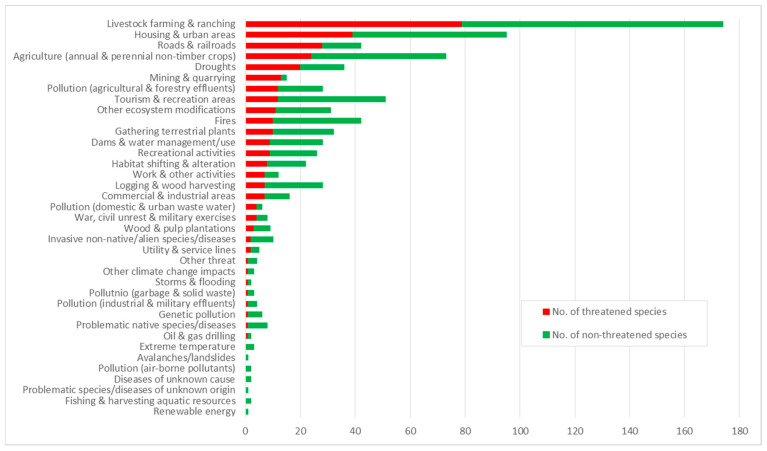
Main threats to CWRs in WANA countries.

**Figure 10 plants-13-01343-f010:**
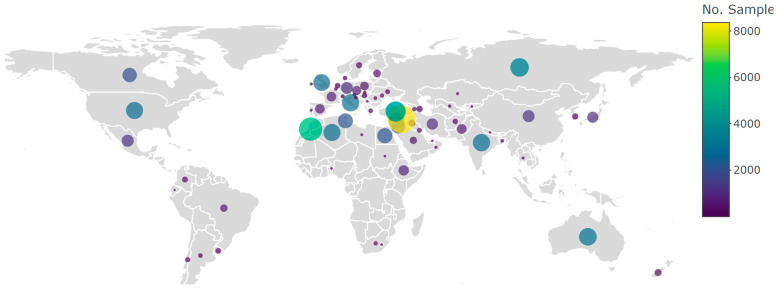
Number of distributed CWR (originated in the WANA region) accessions from the ICARDA collection as per the country of the requester.

**Figure 11 plants-13-01343-f011:**
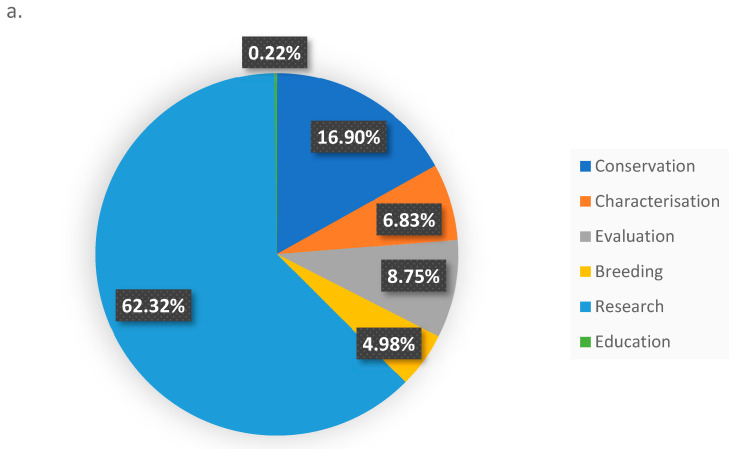
Percentage of samples distributed from the ICARDA CWR germplasm collection that originated in the WANA region (**a**) by use, (**b**) by user type.

**Figure 12 plants-13-01343-f012:**
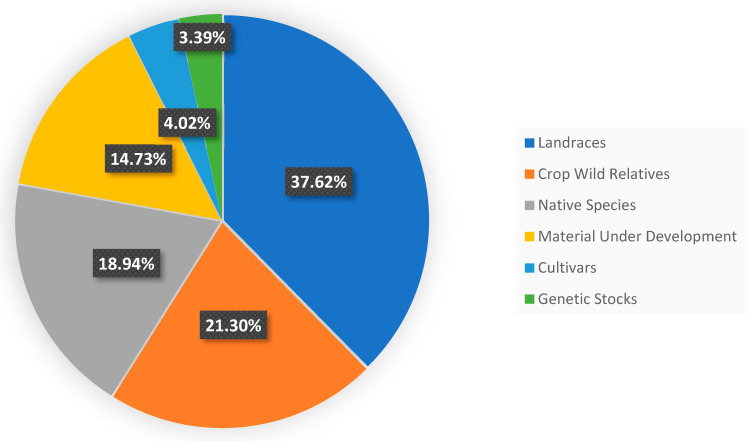
Major components of ICARDA germplasm collections. Approximately 40% of the whole ICARDA collection (ca. 152,000 accessions) are CWRs (33,431) and native species of forage legumes (29,719).

**Table 1 plants-13-01343-t001:** Threatened CWRs in the WANA region. Priority CWRs in the WANA region identified with an asterisk (*) (source: IUCN Red List of Threatened Species) [44]. [Critically Endangered (CR), Endangered (EN), Vulnerable (VU)].

Species	Red List Category	Level of Assessment
*Allium baytopiorum*	CR	Global
*Allium czelghauricum*	CR	Global
*Allium marathasicum*	CR	Global and Mediterranean
*Astragalus acmophylloides*	CR	Global
*Astragalus eliasianus*	CR	Global
*Astragalus kurnet-es-saudae*	CR	Global and Mediterranean
*Astragalus longivexillatus*	CR	Global
*Astragalus nigrocalycinus*	CR	Global
*Astragalus olurensis*	CR	Global
*Astragalus tatlii*	CR	Global
*Astragalus trabzonicus*	CR	Global
*Crataegus turcicus*	CR	Global
*Lathyrus belinensis*	CR	Global and Mediterranean
*Lathyrus gloeosperma*	CR	Global
*Lathyrus phaselitanus*	CR	Global
*Linum carnosulum*	CR	Global and Mediterranean
*Lotus armeniacus*	CR	Global
*Lotus benoistii **	CR	Global, Mediterranean, Northern Africa and Pan-Africa
*Panicum socotranum*	CR	Global
*Rumex tunetanus **	CR	Global, Mediterranean and Pan-Africa
*Salvia peyronii*	CR	Global and Mediterranean
*Salvia veneris*	CR	Global, Europe and Mediterranean
*Vicia eristalioides **	CR	Global
*Vicia erzurumica*	CR	Global
*Vicia fulgens*	CR	Global, Northern Africa and Pan-Africa
*Vicia quadrijuga*	CR	Global
*Allium basalticum*	EN	Global and Mediterranean
*Allium duriaeanum*	EN	Global and Mediterranean
*Allium meronense*	EN	Global and Mediterranean
*Allium noeanum*	EN	Global
*Allium peroninianum*	EN	Global
*Allium pseudoalbidum*	EN	Global
*Allium pseudocalyptratum*	EN	Global and Mediterranean
*Allium sannineum*	EN	Global and Mediterranean
*Allium therinanthum*	EN	Global and Mediterranean
*Amblyopyrum muticum*	EN	Global
*Astragalus acetabulosus*	EN	Global
*Astragalus alamliensis*	EN	Global
*Astragalus cedreti*	EN	Global and Mediterranean
*Astragalus ehdenensis*	EN	Global and Mediterranean
*Astragalus hirsutissimus*	EN	Global and Mediterranean
*Astragalus lanatus*	EN	Global and Mediterranean
*Astragalus transnominatus*	EN	Global and Mediterranean
*Avena murphyi **	EN	Global and Mediterranean
*Barbarea lutea*	EN	Global
*Barbarea macrocarpa*	EN	Global and Mediterranean
*Brassica hilarionis*	EN	Global and Europe
*Carex fissirostris*	EN	Global, Mediterranean and Pan-Africa
*Carum asinorum*	EN	Global, Mediterranean and Pan-Africa
*Cicer bijugum **	EN	Global
*Crocus aerius*	EN	Global
*Cytisus syriacus*	EN	Global and Mediterranean
*Festuca pontica*	EN	Global
*Festuca xenophontis*	EN	Global
*Lathyrus armenus*	EN	Global
*Lathyrus cilicicus*	EN	Global
*Lathyrus hirticarpus*	EN	Global
*Lathyrus layardii*	EN	Global
*Lathyrus libani*	EN	Global
*Lathyrus stenolobus*	EN	Global
*Lathyrus tauricola*	EN	Global
*Lathyrus trachycarpus*	EN	Global
*Lathyrus undulatus*	EN	Global
*Papaver libanoticum*	EN	Global and Mediterranean
*Prunus agrestis*	EN	Global and Mediterranean
*Prunus boissieri*	EN	Global
*Rumex algeriensis**	EN	Global, Mediterranean and Pan-Africa
*Rumex bithynicus*	EN	Global
*Salvia taraxacifolia*	EN	Global
*Trifolium sannineum*	EN	Global and Mediterranean
*Vicia hyaeniscyamus **	EN	Global
*Vicia incisa*	EN	Global
*Vicia kalakhensis **	EN	Global
*Aegilops sharonensis **	VU	Global and Mediterranean
*Allium exaltatum*	VU	Global, Europe and Mediterranean
*Allium pseudophanerantherum*	VU	Global and Mediterranean
*Allium scaberrimum*	VU	Global
*Allium trichocnemis*	VU	Global and Mediterranean
*Astragalus angulosus*	VU	Global and Mediterranean
*Carum lacuum **	VU	Global, Mediterranean and Pan-Africa
*Crocus cyprius*	VU	Global, Europe and Mediterranean
*Crocus hartmannianus*	VU	Global, Europe and Mediterranean
*Cyperus cyprius*	VU	Global, Europe and Mediterranean
*Daucus mirabilis*	VU	Global
*Festuca yemenensis*	VU	Global
*Isatis glastifolia*	VU	Global
*Lactuca tetrantha*	VU	Global and Europe
*Lathyrus cyaneus*	VU	Global
*Lathyrus lycicus*	VU	Global
*Lathyrus nivalis*	VU	Global
*Lathyrus satdaghensis*	VU	Global
*Lathyrus tukhtensis*	VU	Global
*Lathyrus variabilis*	VU	Global
*Leopoldia maritima*	VU	Global and Mediterranean
*Lepidium violaceum*	VU	Global, Mediterranean and Pan-Africa
*Lotus mollis*	VU	Global
*Mentha gattefossei*	VU	Global, Mediterranean and Pan-Africa
*Origanum cordifolium*	VU	Global and Europe
*Origanum ehrenbergii*	VU	Global and Mediterranean
*Origanum libanoticum*	VU	Global and Mediterranean
*Pimpinella lazica*	VU	Global
*Prunus korshinskyi*	VU	Global
*Punica protopunica*	VU	Global
*Pyrus serikensis*	VU	Global
*Rorippa hayanica **	VU	Global, Mediterranean and Pan-Africa
*Vicia esdraelonensis*	VU	Global
*Vicia tigridis*	VU	Global

## Data Availability

Data are contained within the article and Appendix A.

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
