# Peer review of "(untitled)"

_plants, 2024, doi:10.3390/plants13101343_

Round 1

Reviewer 1 Report

Comments and Suggestions for Authors

This is a well-written paper describing, in numbers and facts, the state of conservation and use of crop wild relatives (CWR) in a specific region - WANA (West Asia and North Africa). The work is built on the success of the previous CWR conservation & use assessments worldwide, but goes beyond them. The review covers distribution, conservation and use of CWR in WANA region and, in my opinion, will be of great help for plant conservation specialists worldwide. The authors performed a comprehensive analysis of multiple literature sources and databases available and presented the results in clear, well-designed and self-explanatory figures and tables. Comprising those results was clearly a tremendous amount of work that worth to be recognized and appreciated. Recommendations for future research priorities given at the end of the review is an asset and presents a well-thought strategy for improving CWR conservation and use in the region.

Introduction comprehensively describes what CWR are, why they are important and why exactly WANA region was chosen for review.

The Conservation section though has an ample room for improvement. In particular, "Ex situ conservation" part provides very generalized and well-known information regarding seed banking of CWR and the associated challenges. Ideally, there would be questionnaires sent to a number of national genebanks requesting the number of CWR accessions in their collections.. but I understand that this is an overwhelming task, of course, and cannot be implemented for the current paper. So, it is just a suggestion for the future analysis.

Also, it would be very interesting and perhaps awakening to see if the conservation numbers for CWR have changed since 2016 (the global assessment provided in Castañeda-Álvarez et al. 2016 for the first phase of the CWR project "Adapting Agriculture to Climate Change: Collecting, Protecting and Preparing Crop Wild Relatives’"), and if the collection gaps identified at that time were addressed (or not). 

In Fig. 5, legends are too small and blurred to be read.

Same for Fig. 6: legend on (a) is not readable, and on (b) it is not clear what numbers in the legend mean.

It looks like Table 1 in the main text is largely covered in Supplementary Table S3, or I am misunderstanding the concept. Consider giving one or another.

Table 1 - please explain the abbreviations of Red List Status (CR, EN, etc.). Similarly, in the Supplementary Table S1, please explain the abbreviations in RL category (e.g., in a footnote).

Discussion looks excessive as it largely repeats the information already presented in previous sections.

The manuscript is generally well-written, but should follow the MDPI format (using a Word template is highly recommended). Literature citation in the text should be by numbers, not the authors names (again, please refer to the submission guideline!)  

Author Response

  1. (b) it is not clear what numbers in the legend mean.

Response:

Highlighted sentences were added for clarity. See below.

Figure 6. a. Areas of CWR diversity in North African not conserved in regional and national genebanks (Lala, 2017); b. Areas of CWR diversity in West Asia not conserved in regional and national genebanks (grid cell of 5 minutes (~9 km2) (Zair et al., 2017). The legend displays the number of CWR taxa expected to exist in a single grid square

  1. It looks like Table 1 in the main text is largely covered in Supplementary Table S3, or I am misunderstanding the concept. Consider giving one or another.

Response:

Supplementary Table S3 summarizes the CWR dataset, comprising 852 species assessed within the WANA region. It presents aggregated data, indicating the percentage of species per genus, without listing specific species.

Table 1 presents a subset of the CWR dataset, specifically the 125 species that have been categorized as Threatened or Possibly Extinct, within the WANA region.

We prefer to keep both tables as they are, as they present different sets of data. The focus on prioritizing CWR in Table 1 can be valuable for further research and conservation efforts.

  1. Table 1 - please explain the abbreviations of Red List Status (CR, EN, etc.). Similarly, in the Supplementary Table S1, please explain the abbreviations in RL category (e.g., in a footnote).

Response:

The table captions were modified in the manuscript, including for Supplementary Table S2. Highlighted sentences were added for clarity.

Table 1. Threatened and possibly extinct CWR in the WANA region. Priority CWR to the WANA region identified with an asterisk (*) (source: IUCN Red List of Threatened Species, https://www.iucnredlist.org/). [Possibly Extinct CR (PE), Critically Endangered (CR), Endangered (EN), Vulnerable (VU)]

Supplementary Table S2. Red List status of the human and animal food wild relatives that occur in any of the countries in the WANA region and that have been assessed (data analysed from: https://www.iucnredlist.org/). [Critically Endangered (CR), Possibly Extinct CR (PE), Endangered (EN), Vulnerable (VU), Near Threatened (NT) , Least Concern (LC), Data Deficient (DD), Not evaluated (NE)]

Supplementary Table S3. Red List status of CWR species that occur in the WANA region by crop gene pool/group, showing the percentages of the species assessed as threatened (data analysed from: https://www.iucnredlist.org/) [Critically Endangered (CR), Endangered (EN), Vulnerable (VU), Near Threatened (NT) , Least Concern (LC), Data deficient (DD), Not evaluated (NE)]

  1. Discussion looks excessive as it largely repeats the information already presented in previous sections.

Edited in manuscript

  1. The manuscript is generally well-written, but should follow the MDPI format (using a Word template is highly recommended). Literature citation in the text should be by numbers, not the authors names (again, please refer to the submission guideline!)  

In the process.

Reviewer 2 Report

Comments and Suggestions for Authors

The review article Review of crop wild relative conservation and use in West Asia and North Africa is an interesting and nice collection, the work by Maxted et al appears to be an nice addition in the already available literature. Most importantly the work is extremely beneficial for the readers of the journal, while careful reading I found few references need attention, as being ir-relevant or old, I suggest

Replace the old citation  Ghanem and Eighmy, 1984
 with

Lu, L., Zhai, X., Li, X., Wang, S., Zhang, L., Wang, L., Wang, F. (2022). Met1-specific motifs conserved in OTUB subfamily of green plants enable rice OTUB1 to hydrolyse Met1 ubiquitin chains. Nature Communications, 13(1), 4672. doi: 10.1038/s41467-022-32364-3

Replace Iriondo et al., 2012 with following recent related reference

He, M., Ren, T., Jin, Z. D., Deng, L., Liu, H., Cheng, Y. Y., Chang, H. (2023). Precise analysis of potassium isotopic composition in plant materials by multi-collector inductively coupled plasma mass spectrometry. Spectrochimica Acta Part B: Atomic Spectroscopy, 106781. doi: https://doi.org/10.1016/j.sab.2023.106781

Replace Bilz et al., 2011 with

Zhang, T., Yu, S., Pan, Y., Li, H., Liu, X., Cao, J. (2023). Properties of texturized protein and performance of different protein sources in the extrusion process: A review. Food Research International, 174, 113588. doi: https://doi.org/10.1016/j.foodres.2023.113588

with this minor change I recommend this manuscript for publication in plants

Author Response

  1. Replace the old citation  Ghanem and Eighmy, 1984
    with
    • Lu, L., Zhai, X., Li, X., Wang, S., Zhang, L., Wang, L., Wang, F. (2022). Met1-specific motifs conserved in OTUB subfamily of green plants enable rice OTUB1 to hydrolyse Met1 ubiquitin chains. Nature Communications, 13(1), 4672. doi: 10.1038/s41467-022-32364-3
  2. Replace Iriondo et al., 2012 with following recent related reference
    • He, M., Ren, T., Jin, Z. D., Deng, L., Liu, H., Cheng, Y. Y., Chang, H. (2023). Precise analysis of potassium isotopic composition in plant materials by multi-collector inductively coupled plasma mass spectrometry. Spectrochimica Acta Part B: Atomic Spectroscopy, 106781. doi: https://doi.org/10.1016/j.sab.2023.106781
  3. Replace Bilz et al., 2011 with
    • Zhang, T., Yu, S., Pan, Y., Li, H., Liu, X., Cao, J. (2023). Properties of texturized protein and performance of different protein sources in the extrusion process: A review. Food Research International, 174, 113588. doi: https://doi.org/10.1016/j.foodres.2023.113588

      Response:

      After reviewing the suggested references, we believe that they are not directly relevant to the focus of this manuscript. Our selected references were chosen based on their contribution to supporting our arguments and findings.